# Optimization of Extraction Process, Preliminary Characterization and Safety Study of Crude Polysaccharides from Morindae Officinalis Radix

**DOI:** 10.3390/foods12081590

**Published:** 2023-04-09

**Authors:** Yaxian Chen, Yini Cai, Zhimin Zhao, Depo Yang, Xinjun Xu

**Affiliations:** School of Pharmaceutical Sciences, Sun Yat-Sen University, Guangzhou 510006, China

**Keywords:** Morindae officinalis radix, crude polysaccharides, hot water extraction, optimization, preliminary characterization, safety study

## Abstract

In this study, the hot water extraction process of crude polysaccharides from Morindae officinalis radix (cMORP) was conducted and optimized through a single-factor test and orthogonal experimental design. With the optimal extraction process (extraction temperature of 80 °C, extraction time of 2 h, liquid/solid ratio of 15 mL/g, and number of extraction of 1), the cMORP was obtained by the ethanol precipitation method. The chemical properties and preliminary characterization of the cMORP were analyzed by chemical or instrumental methods. Furthermore, to indicate a preliminary study on safety, a single oral dose of 5000 mg/kg body weight (BW) was administered orally to Kunming (KM) mice for acute toxicity, and the cMORP was administered orally to KM mice once a day at doses of 25, 50, and 100 mg/kg BW for 30 days. General behaviors, body weight variations, histopathology, relative organ weights, and hematological and serum biochemical parameters were observed and recorded. The results suggested there were no toxicologically significant changes. Based on the safety study, cMORP can be initially considered non-toxic with no acute oral toxicity up to 5000 mg/kg BW and safe at up to 100 mg/kg BW in KM mice for 30 days.

## 1. Introduction

Morindae officinalis radix (MOR), the dried root of medicinal plant *Morinda officinalis* How (Rubiaceae), has long been used in tonics and nutrient supplements (healthcare products, nourishing soups and drinks) in the southeast region of China for its action in nourishing the kidney, anti-osteoporosis, immune-enhancing effects, and for alleviating a wide spectrum of diseases. Various types of chemical constituents of MOR have been researched and reported, such as polysaccharides, mono- and oligosaccharides, iridoids, glycosides anthraquinones, volatile oils, organic acids, and other types of compounds [1]. Based on that, MOR has been shown to have multiple biological activities, including anti-depression [2,3,4,5], anti-inflammation [6,7], anti-osteoporosis [8,9,10], anti-fatigue [11], anti-Alzheimer’s disease (AD) [12,13,14], anti-oxidation [11,15,16], immune-regulation [17,18], pro-fertility [19,20,21], anti-radiation [21,22], regulation on gut microbiota [15,23,24], etc. The content of saccharides in MOR was reported to be 49.79–58.25% [12], and polysaccharides are the main active substances of MOR. Until now, studies on a variety of biological activities of the total polysaccharides or crude polysaccharides extracted from MOR, such as antioxidant [15,16,25], pro-fertility [26,27], anti-fatigue [11,28], bone protection and anti-osteoporosis [9,29,30,31,32], and anti-inflammatory [33] have been reported, indicating that the polysaccharides have been supposed to play important roles in the pharmacological properties. In addition, more studies have revealed the structure and molecular mechanisms underlying the biological activities.

MOR has been approved since 2002 as a functional food for daily healthcare by the National Health Commission of China and has been used in health foods and Chinese prescriptions, such as *Baoshen Wan*, *Tianjing Bushen Gao*, and *Erxian Decoction* [1]. The development of health-functional foods and new food raw materials using natural products or extracts is actively advancing. Recently, the toxicological characteristics of fermented Morinda officinalis (FMO) have been assessed for supporting the safe use of FMO as a functional food and medicine [34]. Most active extracts derived from herbal medicines are considered safe, but some of them have known toxicities [35,36]. Therefore, the scientific evaluation and basis for the safety of extracts are considered important and essential [37,38,39]. The crude polysaccharides extracted from MOR have diverse pharmacological effects, which indicates great potential for applications in the development of new food raw material or functional food. However, the evaluation of potential toxicity of cMORP has not been reported so far. The evidence suggests that the biological activities are related to structural characteristics of polysaccharides (chemical structure, monosaccharide composition, molecular weight, etc.) affected by the selected methods of extraction [40,41,42]. The extraction methods of polysaccharides are becoming increasingly diverse, including hot water extraction, alkali solution extraction, ultrasound-assisted extraction, microwave-assisted extraction, subcritical water extraction, and so on. It is hoped that the physical properties of polysaccharides will not change significantly under mild extraction conditions. The hot water procedure is usually the preferred method for extracting polysaccharides in industrial production, largely due to the process simplicity, facility, low-cost, and safety. A simple and effective extraction method of polysaccharides is the first step for development and application. Hot water extraction was chosen for the following experiments in view of simplicity and industrialization, and the extraction process should be optimized.

Therefore, in this study, the hot water extraction process of cMORP, including four variables (liquid/solid ratio, extraction temperature, number of extraction times, and extraction time), was investigated and optimized via a single-factor test and orthogonal experimental design. Then, ultraviolet–visible spectroscopy (UV-vis), Fourier-transform infrared spectroscopy (FT-IR), high-performance anion-exchange chromatography (HPAEC), matrix-assisted laser desorption ionization time of flight mass spectrometry (MALDI-TOF-MS), and scanning electron microscopy (SEM) were utilized for the preliminary characterization of cMORP. Additionally, a 14-day acute toxicity study in vivo with a single-dose and 30-day repeated tests at different doses were conducted to establish a preliminary safety evaluation. The aim of this research is to isolate water-soluble crude polysaccharides from Morindae officinalis radix and to evaluate the structural characteristics and safety. The results of this research are expected to provide a scientific basis and support for the practical application of cMORP as a new food raw material or functional food and in-depth study.

## 2. Materials and Methods

### 2.1. Reagents and Animals

MOR was obtained from Deqing County, Zhaoqing City, Guangdong Province, China. After drying and crushing, it was sieved through a 60-mesh sieve and placed in a dryer. Thirteen kinds of standard monosaccharides, including rhamnose (Rha), arabinose (Ara), glucose (Glc), galactose (Gal), fucose (Fuc), xylose (Xyl), mannose (Man), fructose (Fru), ribose (Rib), galacturonic acid (Gal-UA), glucuronic acid (Glc-UA), mannuronic acid (Man-UA), and guluronic acid (Gul-UA) were purchased from Aladdin Reagent Co., Ltd. (Shanghai, China). Other chemicals and reagents were of analytical grade.

The specific pathogen free (SPF) KM mice (half male and half female) were obtained from the Laboratory Animal Center of Sun Yat-Sen University (Guangzhou, China), and the license was SCXK (Y) 2021-0029. Adult healthy KM mice aged 6–8 weeks were used for the acute toxicity test (body weight, 25–30 g (males); 20–25 g (females)). Adult healthy KM mice aged 6–8 weeks were used for the repeated-dose 30-day oral toxicity study (body weight, 25–30 g (males); 25–30 g (females)). The selected female mice were non-pregnant and nulliparous. Male and female mice were housed separately and given sterilized food (Guangdong Medical Laboratory Animal Center, Guangdong, China) and purified water via a water bottle. After three days of acclimatization feeding, the mice were randomly assigned to different cages and labeled, and the temperature of the feeding environment was controlled at 22 ± 3 °C. The animal study was performed in accordance with the protocol approved by the Animal Ethical and Welfare Committee of Sun Yat-sen University (Approval No. SYSU-IACUC-2023-000015).

### 2.2. Optimal Design of Hot Water Extraction Process for cMORP

#### 2.2.1. Hot Water Extraction Process

The cMORP was extracted by a hot water procedure. The dried powder was added to water at a ratio of 1:20 (*w*/*v*) and extracted twice at 80 °C for 2 h each time, and the residue was filtered. The polysaccharide extract solution was obtained, and the content of polysaccharides was determined by the phenol-sulfuric acid method [43]. The calculation formula of yield of polysaccharides was as follows:(1)$$\mathrm{Extraction}\;\mathrm{yield}(\%)=\frac{C\times V\times D}{M}\times 100\%$$
where *C*, *V*, *D*, and *M* are the concentration of polysaccharides (g/mL), volume of the solution (mL), dilution factor, and the mass of the sample (g), respectively.

#### 2.2.2. Single-Factor Experiment

The medium value of the optimal range for each factor was determined by single-factor experiments. The values of other three factors remained fixed when one factor changed. The yield of polysaccharide was affected by multiple factors, including extraction temperature, extraction time, liquid/solid ratio, and extraction times [44]. The four factors, including liquid/solid ratios of 10, 15, 20, 25, and 30 mL/g; extraction temperatures of 60, 70, 80, 90, and 100 °C; numbers of extraction times of 1, 2, 3, 4, and 5; and extraction times of 0.5, 1, 2, 3, and 4 h, were set as the experimental conditions and investigated.

#### 2.2.3. Orthogonal Experimental Design

Based on a preliminary study of extraction factors (liquid/solid ratio, extraction temperature, number of extraction times, and extraction time) in the single-factor trials, the orthogonal table L_9_(3^4^) was selected to optimize the parameters, reflecting the orthogonality, representativeness, and comprehensive comparability. The assessment indicator was the yield of polysaccharides, and then the optimal extraction process conditions were determined. Table 1 shows the test factors and levels of the experiments.

#### 2.2.4. Obtention of cMORP

The extraction process was carried out according to the optimized method. The extract was concentrated to one-third of the volume by a rotary evaporator, decolorized twice with 1% activated carbon (*w*/*v*), and filtered. The supernatants were recovered and concentrated to a quarter of volume, and the polysaccharides extract solution was obtained. cMORP was obtained by the alcohol precipitation method. Subsequently, the concentrated extract was mixed with 95% ethanol to an ethanol concentration of 80%, stirred, and kept overnight. The precipitate was collected by centrifugation (15 min at 5000 rpm). The precipitate was dried under vacuum and stored in a light-proof desiccator.

### 2.3. Characterization of cMORP

#### 2.3.1. Determination of Chemical Composition

The polysaccharide content was determined by the phenol-sulfuric acid method with glucose as the standard [45]. In short, 1 mL polysaccharide solution or glucose solution at a concentration of 50 μg/mL was mixed with 1 mL 5% phenol solution and 5 mL sulfuric acid and then reacted at 100 °C for 15 min. After cooling to room temperature, the absorbance at 490 nm was measured. The uronic acid content was determined by the vitriol-carbazole method with D-glucuronic acid as the standard [46]. Briefly, 1 mL polysaccharide solution or D-glucuronic acid solution was mixed with 6 mL sodium tetraborate-sulfuric acid solution in an ice-water bath and then reacted at 100 °C for 10 min. After cooling to room temperature, it was mixed with 100 μL of m-hydroxybiphenyl solution with the absorbance measured at 524 nm. The protein content of cMORP was determined using Coomassie brilliant blue G-250 with bovine serum albumin (BSA) as the standard [47]. In short, 1 mL polysaccharide solution or BSA solution was mixed with 1 mL Coomassie brilliant blue G-250 solution and then reacted at room temperature for 30 min with absorbance measured at 590 nm.

#### 2.3.2. UV-Vis and FT-IR

For the analysis, 1 mg/mL aqueous solution of cMORP was prepared and scanned from 200 to 800 nm at room temperature with a UV-2600 ultraviolet spectrophotometer (Shimadzu, Japan), and the UV-vis spectrum was obtained. cMORP functional groups were determined by FT-IR. Then 10 mg of the sample was pressed into the disk. The IR spectrum was recorded at room temperature (25 ± 0.5 °C) with a spectral resolution of 2 cm^−1^ under dry air in the range of 4000–450 cm^−1^ on an FT-IR spectrophotometer (Frontier, PerkinElmer, Waltham, MA, USA) [48].

#### 2.3.3. Determination of Molecular Weight (Mw)

MALDI-TOF-MS is sensitive to the detection of low molecular weight substances and can determine the molecular mass distribution of crude polysaccharides. The sample were diluted with distilled water to a concentration of 1 mg/mL and filtered. The signals were recorded in the range of 400–4000 m/z.

#### 2.3.4. Monosaccharide Identification

The monosaccharide composition was determined by HPAEC. The polysaccharides were hydrolyzed in a sealed chromatographic bottle with 2 M trifluoroacetic acid (TFA) for 2 h at 121 °C and redissolved with water and analyzed by HPAEC following the removal of TFA. The retention times and standard curves of monosaccharide standards, including Rha, Ara, Glc, Gal, Fuc, Xyl, Man, Fru, Rib, Gal-UA, Glc-UA, Man-UA, and Gul-UA, were determined. By comparing the chromatographic peak and retention time of the sample with that of the standards, the composition and content of the monosaccharide in the sample can be determined.

Chromatographic conditions were as follows: Dionex™ CarboPac™ PA-20 column (3.0 mm × 150 mm, 10 µm); Dionex™ ICS-5000 system with pulsed amperometric detector (PAD); flow rate: 0.5 mL/min; column temperature: 30 °C; injection volume: 5 µL; mobile phase: ddH_2_O (A); 0.1 M NaOH (B); 0.1 M NaOH, 0.2 M NaAc (C); gradient elution (0–26 min, 95–85% A, 5–5% B, 0–10% C; 26–42 min, 85% A, 5% B, 10% C; 42–42.1 min, 85–60% A, 5–0% B, 10–40% C; 42.1–52 min, 60–60% A, 0–40% B, 40–0% C).

#### 2.3.5. SEM Analysis

The dried sample was sieved through a 100-mesh sieve and sputter-coated with a gold layer. The surface structural characteristics of cMORP were examined via a scanning electron microscope (Merlin Compact, Zeiss, Germany), and then the images were recorded at different magnifications (250–2000 times) at an acceleration voltage of 3 kV.

### 2.4. Safety Study

#### 2.4.1. Single-Dose Acute Oral Toxicity Study

The experiments were performed according to Technical Guidelines for Acute Toxicity Research of Traditional Chinese Medicine and Natural Medicine. For acute oral toxicity, healthy adult mammals of half female sex were generally used. A total of 40 mice with half male and half female was randomly divided into the control group (0 mg/kg) and test group (5000 mg/kg). The samples were diluted with distilled water. The negative control group was administered with distilled water. After a 3-day acclimatization period, body weights were measured before administration (10/sex/group). After a 12 h fast, a single oral dose of cMORP or distilled water was administered separately by an oral gavage needle (40 mL/kg). Food was provided 4 h after the administration. Furthermore, signs of toxicity and behavioral changes were observed carefully within 4 h after administration, including water consumption, appearance, behavior, secretions, excreta, etc. The signs of clinical symptoms and mortality of experimental animals were observed for 14 days thereafter. Additionally, the body weights were measured on days 0, 7, and 14. The day of administration was designated as day 0. On the 14th day after administration, all mice were fasted for 16 h and then euthanized. The major organs, including the heart, liver, spleen, lung, kidneys, and testes, were visually examined following animal autopsies and collected for pathological section observation. 

#### 2.4.2. Repeated Dose 30-Day Oral Toxicity Study

After a 3-day acclimatization feeding, 80 mice, half male and half female, were divided randomly into four groups, namely, a low-dose group, a medium-dose group, and a high-dose group (0, 25, 50, and 100 mg/kg, respectively), as well as a negative control group. The samples were diluted with distilled water. The negative control group was administered with distilled water. An oral dose of cMORP or distilled water was administered separately by an oral gavage needle (40 mL/kg) once a day. The signs of clinical symptoms and mortality of experimental animals were observed twice a day for 30 days thereafter. The weights of each animal were recorded on days 0, 7, 14 and 28. The day of administration was designated as day 0.

After 30 days, animals were fasted for 16 h and then euthanized. Blood samples were collected for hematological examination and serological analysis. The major organs, including the heart, liver, spleen, lung, kidneys, and testes, were visually examined following animal autopsies, weighed, calculated as the relative weight, and collected for histological examination.

### 2.5. Statistical Analyses

The experimental data are represented as the mean ± standard deviation (SD). Where appropriate, one-way analysis of variance (ANOVA) with LSD’s post test was used to assess the differences between multiple groups. Statistical analysis was conducted using SPSS version 22. In this study, the significance level was set at *p* < 0.05. The data were processed and obtained by the software GraphPad Prism 5.

## 3. Results and Discussion

### 3.1. Single Factor Experiments Assessment

The effect of the liquid/solid ratio on polysaccharide yield is shown in Figure 1A. The yield of polysaccharides tended to increase until the liquid/solid ratio reached 15 mL/g, but after exceeding 15 mL/g, it no longer increased significantly. The extraction of polysaccharides was incomplete when the liquid/solid ratio was relatively low. The extraction system reached a saturated state with an increase in the liquid/solid ratio. Consequently, within a certain range, the contact of the material–liquid phase was gradually sufficient, and the viscosity of the extraction solution decreased with the increase of the liquid/solid ratio, leading to an increase in the yield of polysaccharides. However, an excessive liquid/solid ratio can lead to a waste of solvent and is not conducive to subsequent concentration processes.

The diffusion coefficient was positively correlated with temperature. The higher diffusion rate could be obtained by increasing the extraction temperature. The permeability of plant cell membranes, and the leaching and solubility of carbohydrate in solvent can increase and be facilitated with the increasing of temperature. As shown in Figure 1B, the yield increased noticeably and peaked at 80 °C as temperature was raised. The yield was then reduced, possibly because of excessive temperature, causing the degradation and inactivation of the carbohydrate. Thus, the optimal extraction temperature was 80 °C.

The number of extraction times can affect the yield and cost of the process, especially at an industrial scale. Therefore, it is one of the important parameters in an investigation of extraction processes. As seen in Figure 1C, the yield of polysaccharides increased with an increase in number of extraction times from 1 to 2, and beyond that level, it was constant and stable.

In general, longer extraction time was beneficial for increasing the yield of the target substance, but it may lead to decomposition and structural destruction. The yield of polysaccharides noticeably increased with the extension of extraction time and reached the maximum value at 2 h, but it tended to decrease gradually when the time of extraction exceeded 2 h (Figure 1D). This may be due to the reason mentioned above.

### 3.2. Orthogonal Experimental Design

Orthogonal experimental design, an experimental design method to study multi-factors and multi-levels, was used to replace the full-scale test. Relying on the orthogonality of the orthogonal table to select some representative points from the full-scale test, it can achieve equivalent results to a large number of full-scale tests with a minimum number of tests. It can reduce the number of tests. In this test, an orthogonal experimental design was used to obtain the optimal combination scheme quickly. The main order of the influence of each factor on the test index was judged by the size of the comparison of the range, and then the optimal conditions were determined. Table 2 shows the order to be R_B_ > R_D_ > R_C_ > R_A_. Therefore, the main and secondary orders of factors on the yield of polysaccharides were extraction temperature, time, number of times, and liquid/solid ratio, and the optimal extraction process was A_1_B_2_C_1_D_3_, as follows: extraction temperature of 80 °C, extraction time of 2 h, liquid/solid ratio of 15 mL/g, and number of extraction times of 1.

### 3.3. Analysis

The experiments were carried out in triplicate under optimal extraction conditions to compare the experimental values with predicted results. Table 3 shows the experimental yield of polysaccharides under optimal conditions, namely, 8.37 ± 2.44%, which was close to the highest value in the orthogonal experiment (8.52%), indicating that the optimal extraction process conditions were determined through the orthogonal experiment.

### 3.4. Characterization

#### 3.4.1. Chemical Properties Analysis

In Table 4, the content of polysaccharides in cMORP was determined to be 99.80 ± 1.52% by the phenol-sulfuric acid assay. The contents of uronic acid and protein were each less than 0.01%.

#### 3.4.2. UV-Vis and FT-IR

In the UV-vis spectrum, there was no significant absorption at 260 and 280 nm (Figure 2A), indicating the absence of nucleic acid and protein in cMORP, respectively [49]. It was consistent with the previous chemical properties analysis.

FT-IR, an important tool for analyzing and identifying the linkage bonds and structures in polysaccharides, can record the characteristic absorption peaks of distinct functional groups. The FT-IR spectrum of cMORP is shown in Figure 2B. There was a broad absorption peak at 3271.34 cm^−1^ corresponding to the stretching vibration of the O–H bond [50]. A weak absorption peak at 2928.69 cm^−1^ was caused by the stretching vibration of the C–H bond [51]. The signal at 1637.01 cm^−1^ could be associated with the stretching vibration of the C–O group [52] and related to the asymmetric stretching vibration of the C=O bond. The absorption peak around 1431.93 cm^−1^ was attributed to the angular vibration of the C–H bond [53]. The signals around 1200–1000 cm^−1^ were related with the C–O–H stretching vibration and C–O–C glycosidic bond vibrations [49], indicating the presence of pyran rings. The weak absorption peaks at 800–900 cm^−1^ were attributed to the α- and β-configurations [54]. In summary, cMORP exhibited typical polysaccharide characteristics.

#### 3.4.3. Determination of Molecular Weight

The molecular weight of cMORP was measured and obtained by MALDI-TOF mass spectrometry (Figure 3). The molecular weight distribution and subtle differences of polysaccharides could be visualized from the spectrogram, and the type of polysaccharides could be calculated from the accurate ion peaks, providing an ideal analytical method for further analysis of the chemical structure of polysaccharides. MALDI-TOF-MS can allow for the determination of the full molecular mass distribution of a polymer and gives absolute molecular masses rather than relative values. As seen in Figure 3B, there was a cluster of peaks with 162 Da differences in the range of 689.22–2796.85 Da. The polysaccharides in the sample consisted of 10–17 monosaccharides.

#### 3.4.4. Monosaccharide Identification

Crude polysaccharides are mixtures that contain many species of monosaccharides after hydrolysis. Determination of the species and content of monosaccharides is important to further explore the structure and activity of polysaccharides. Monosaccharide compositions were determined by HPAEC. The HPAEC spectra of monosaccharide standards and cMORP are shown in Figure 4. The species and contents of monosaccharides in cMORP are shown in Table 5. The results demonstrated that cMORP was composed of Ara, Gal, Glc, Rha, Gal-UA, Glc-UA, and Gul-UA [55,56]. cMORP mainly consisted of Glc (92.15%), which provided a basis for further exploration of structure and activity.

#### 3.4.5. SEM Analysis

Scanning electron microscopy (Figure 5) can provide information for characterizing microscopic appearance and analyzing structural morphology. The surface morphology of the polysaccharide was visually observed at different magnifications. As seen in Figure 5A, cMORP consisted of irregular pieces and particles with rough surfaces that were observed at low magnification. At a magnification of 2000 times (Figure 5D), the surfaces of particles were shown to be irregular and rough, with pores of different sizes, showing an uneven distribution. There were differences in surface morphologies of polysaccharides, which may be related to the angle of observation, structure, and physicochemical properties of the polysaccharides, extraction type, drying method, and so on.

### 3.5. Safety Study

#### 3.5.1. Single-Dose Acute Oral Toxicity Study

Based on the results of the acute oral toxicity study, we can make a preliminary assessment of the appropriate dose and toxicity of the test substance. The test substance was administered orally to KM mice one time. Within four hours of administration, the mice drank more water, which was caused by the oral administration of high concentrations of the test substance. Moreover, there were no abnormal changes in eyes or mucous membranes of the mice, nor were there any abnormalities in body movements, respiratory patterns, or behavior. The presence of convulsions, drooling, diarrhea, sleepiness, or other pathological signs were also not observed. The mice were observed daily for 14 days after administration for general signs and mortality. No mortality, changes in gross appearance, abnormalities in behavior, and pathological signs were observed in the 14 days after administration. Body weight was measured before and 7 and 14 days after administration. There was no significant difference in body weight gain in KM mice between the treated and control groups of the same sex (*p* > 0.05) (Figure 6) before, during, and after the experiment.

After 14 days, autopsies were performed on the major organs of all 16 h-fasted KM mice. No pathological changes were observed in all KM mice (data not shown). The heart, the vital organ to maintain the characteristics of life; the kidneys, spleen, and liver, the vital immune organs for metabolism; the lung, an important component of the respiratory system; the intestine, the first organ exposed to the test substance; and the testes, the potential organ exposed to the test substance, were sectioned, embedded, and stained with hematoxylin and eosin (HE) for histological examination (Figure 7). It can be concluded that there were no gross abnormalities or histopathological changes in the liver, spleen, kidneys, large intestine, and testes, and there were no significant differences between the treated groups and the control group of the same sex. The cardiac myocytes were irregularly short and cylindrical, with branches that interconnected into a network. In the treated groups and the control group of the same sex, the hearts and lungs of the experimental animals showed partial bruising, which was due to improper handling and squeezing during blood sampling. The central vein was located in the center of the hepatic lobule. The well-defined hepatocytes surrounding it were arranged in a radial pattern. The cells were closely packed with abundant cytoplasm and uniform color. The structure of hepatic lobules was intact without abnormal space enlargement, and the liver tissue was free of inflammation. Overall, the shape and structure of the renal tubules, renal corpuscles, and capillary bulbs were normal with no obvious granular degeneration, interstitial congestion, intracellular hyperplasia, or inflammatory cell infiltration. The spleen showed no amyloidosis, necrosis, or inflammation. The intestinal mucosa was in a normal shape and structure without mucosal hyperplasia, and the crypt was visible [38]. The lymphocytes and plasma cells in the lamina propria had normal morphology and were not detached or separated. The spaces between the seminiferous tubules were clearly visible without shed cell masses, and no pathological shedding of seminiferous epithelial cells occurred. All these features could lead us to the conclusion that there was no necrosis, fibrosis, or loss of normal structure in the major organs in the HE-stained tissue sections. In the single-dose acute oral toxicity test, there were no mortality or clinical changes of toxicity, and the median lethal dose (LD_50_) of cMORP was estimated to be more than 5000 mg/kg BW. The test substance could be initially considered to be non-toxic [57].

#### 3.5.2. Repeated Dose 30-Day Oral Toxicity Study

A 30-day repeated oral toxicity study was performed to assess the potential hazards to the health of the organism that may be caused by repeated exposure to the test substance and to provide information on target organ and accumulation toxicity. In toxicity studies, body weight, organ index, HE-stained tissue sections, and hematological and serum biochemical parameters are considered as important indicators of the adverse or deleterious effects of the test substance.

Change in body weight can directly reflect the overall health status of mice. Body weight was tracked on days 0, 7, 14, and 28 (Figure 8). There was no significant difference in the weight change of the mice in each group of the same sex (*p* > 0.05) in the experiment. At the same time, no mice died or demonstrated unusual clinical signs during the entire administration. At the end of 30 days of feeding, the major organs of experimental animals from the control and the high-dose group were collected and examined histologically. Figure 9 shows that partial bruising occurred in the hearts and lungs of the high-dose group and the control group of the same sex because of improper handling and squeezing. In general, the organs were free of necrosis, fibrosis, inflammation, or loss of normal structure. The histological examination of the major organs showed no differences between the control and high-dose group. The light micrographs of the sections of the heart, liver, spleen, lung, kidneys, large intestine, and testes of mice showed no significant difference between the 30-day repeated-dose oral toxicity study and the acute toxicity study in the histological examination. Moreover, the morphological changes of the major organs were analyzed macroscopically by calculating the ratio of each organ to body weight; relative organ weights are shown in Table 6. The results showed that there was no remarkable difference in the viscera index of heart, liver, spleen, lung, kidneys, and testes among the male and female mice of different groups.

White blood cells (WBCs) consisted of neutrophils, eosinophils, basophils, lymphocytes, and monocytes, protecting the body from allergies, infections, and diseases by mediating the immune system. It played an important role in the healing of body damage, the resistance to invasion of pathogens, and in the mediating of immune systems. The study of hematological parameters can provide a basis for assessment of inflammatory conditions, infectious assessment, and potential damage of the test substance [38]. The hematological parameters are summarized in Table 7, suggesting there was no significant difference in the hematological parameters between the control and test groups (*p* > 0.05). The individual values of hematological parameters in groups were within normal biological variation [58]. The values of alanine aminotransferase (ALT) and aspartate aminotransferase (AST), mainly found in the hepatocyte cytoplasm and mitochondria, respectively, can be significantly increased in the serum when the liver tissue is lesioned and damaged. Moreover, serum concentrations of total protein (TP), albumin (ALB), glucose (GLU), creatinine (CR), cholesterol (CHO), and triglyceride (TG) were measured to evaluate functions of the liver and renal system. The serum biochemical parameters are summarized in Table 8, revealing no statistically significant differences between the control and test groups. The test substance had no significant effect on the regulation of blood glucose concentration and energy metabolism. The repeated oral dose toxicity study for 30 days in KM mice at doses of 0, 25, 50, and 100 mg/kg demonstrated that the test substance was not a trigger for an inflammatory response in KM mice and was considered safe under the experimental conditions of the study.

## 4. Conclusions

In this study, the hot water extraction of crude polysaccharides from Morindae officinalis radix was optimized by the single-factor test and the orthogonal design method. The optimal extraction process involved an extraction temperature of 80 °C, an extraction time of 2 h, a liquid/solid ratio of 15 mL/g, and an extraction time number of 1, and it had an extraction yield of 8.37 ± 2.44%. The preliminary characterization of cMORP demonstrated that it had obvious and characteristic peaks of polysaccharides in the FT-IR spectrum, and the relative molecular masses were mainly distributed in the range of 689.22–2796.85 Da in the MALDI-TOF-MS spectrum. The analysis of monosaccharide composition showed that cMORP primarily comprised glucose (92.15%), guluronic acid (2.28%), galacturonic acid (1.98%), galactose (1.51%), rhamnose (0.88%), arabinose (0.81%), and glucuronic acid (0.40%). Moreover, a single-dose acute oral toxicity test and a repeated-dose 30-day oral toxicity study were performed to evaluate the feasibility of cMORP as a new food raw material. The preliminary study of safety suggested that an LD_50_ of cMORP was estimated to be greater than 5000 mg/kg BW, and oral administration of cMORP at a concentration of 0, 25, 50, and 100 mg/kg presented no abnormal symptoms and adverse toxicological effects, as evidenced by the absence of treatment-related changes in gross appearance, body weight, behavior, histological examination, and hematological and serological parameters of male and female KM mice. Therefore, it was concluded that cMORP can be initially considered non-toxic, and this study can provide valuable information for future development and application.

## Figures and Tables

**Figure 1 foods-12-01590-f001:**
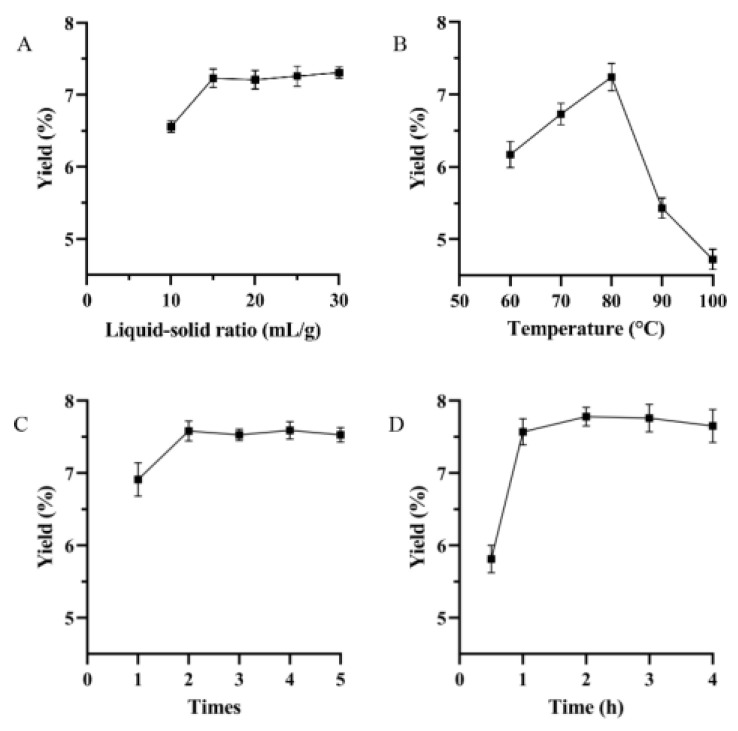
The effect of extraction factors on the yield of polysaccharides (*n* = 3): (**A**) liquid/solid ratio; (**B**) temperature; (**C**) times; (**D**) time.

**Figure 2 foods-12-01590-f002:**
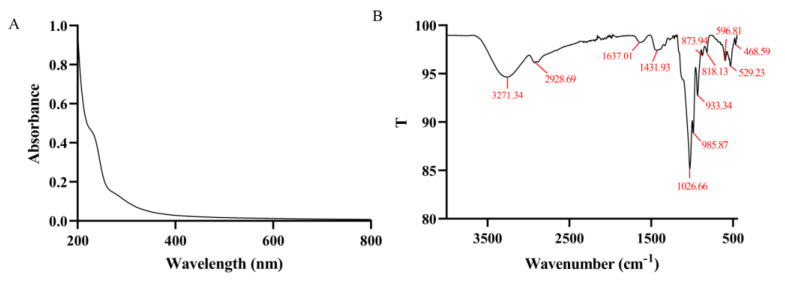
(**A**) UV−vis spectrum of cMORP in the range of 200–800 nm; (**B**) FT−IR spectrum of cMORP in the range of 4000–450 cm^−1^.

**Figure 3 foods-12-01590-f003:**
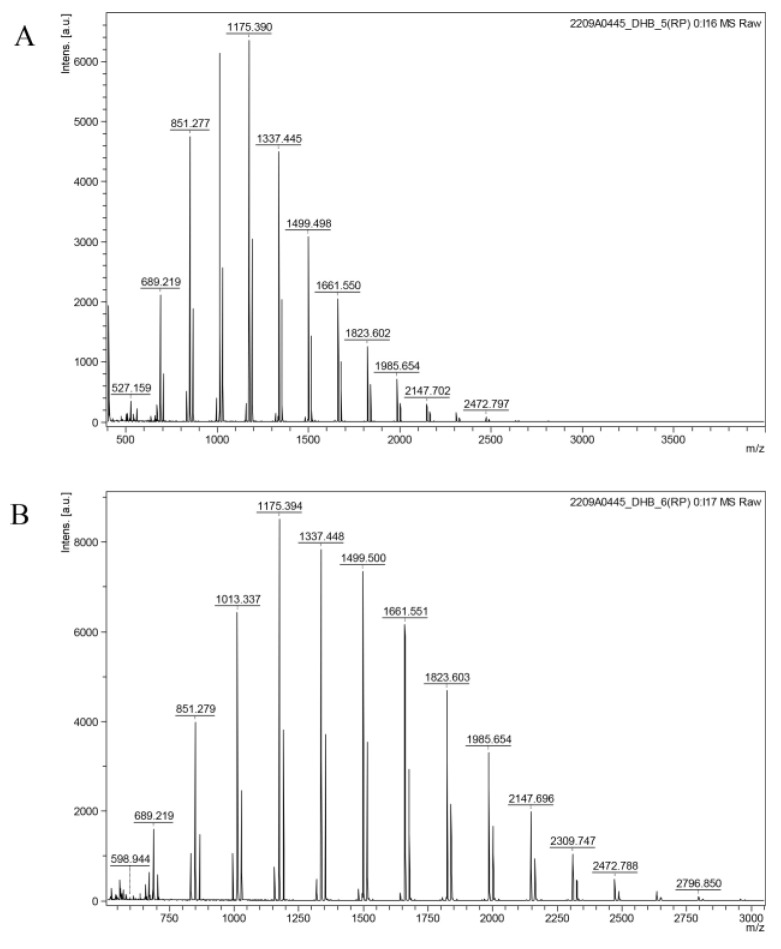
(**A**) MALDI-TOF-MS spectrum of cMORP in the range of 400–4000 m/z; (**B**) MALDI-TOF-MS spectrum of cMORP in the range of 500–3000 m/z.

**Figure 4 foods-12-01590-f004:**
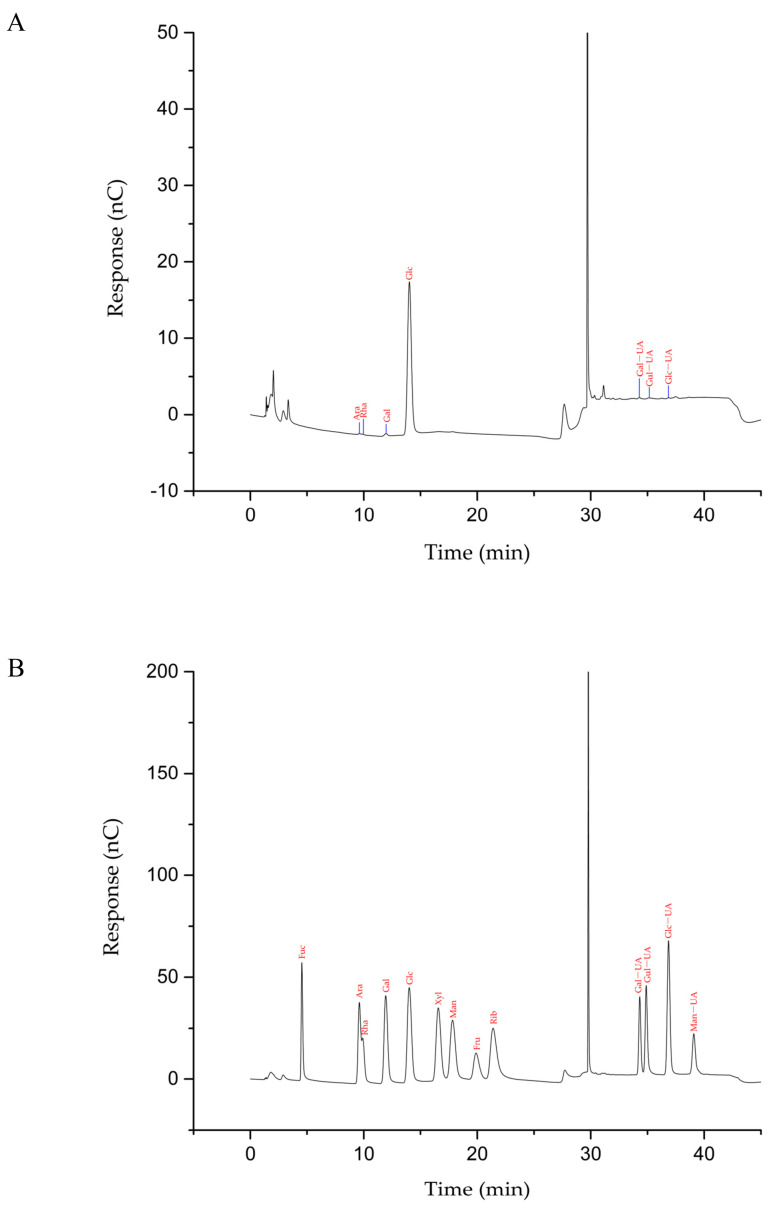
(**A**) The HPAEC chromatogram of cMORP; (**B**) the HPAEC chromatogram of monosaccharide standard.

**Figure 5 foods-12-01590-f005:**
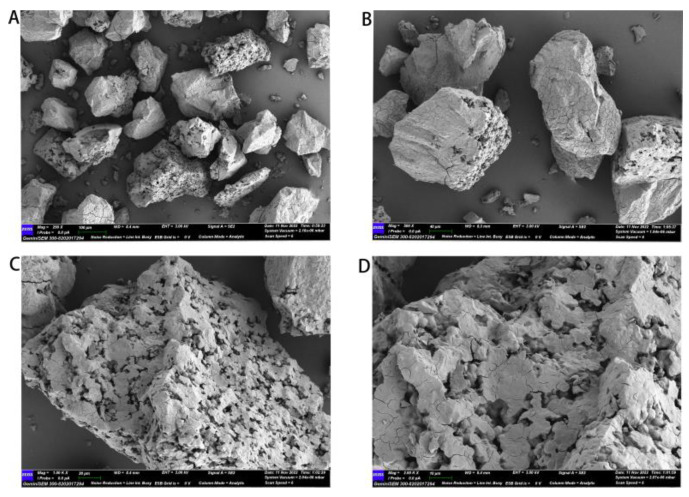
(**A**) SEM images of cMORP ×250; (**B**) SEM images of cMORP ×500; (**C**) SEM images of cMORP ×1000; (**D**) SEM images of cMORP ×2000.

**Figure 6 foods-12-01590-f006:**
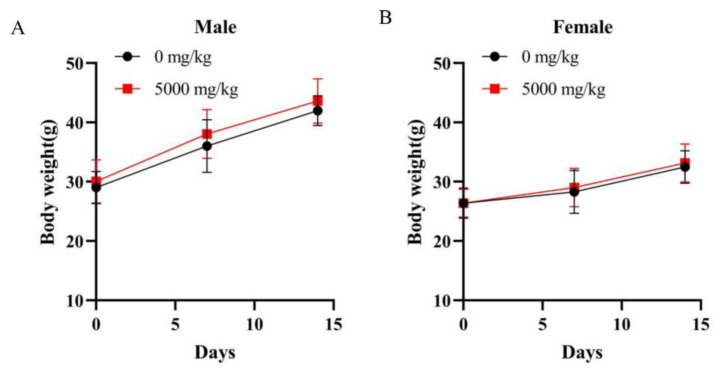
Body weight changes of mice in the single-dose acute oral toxicity study: (**A**) male; (**B**) female. Data are presented as mean ± SD (*n* = 10/sex/group).

**Figure 7 foods-12-01590-f007:**
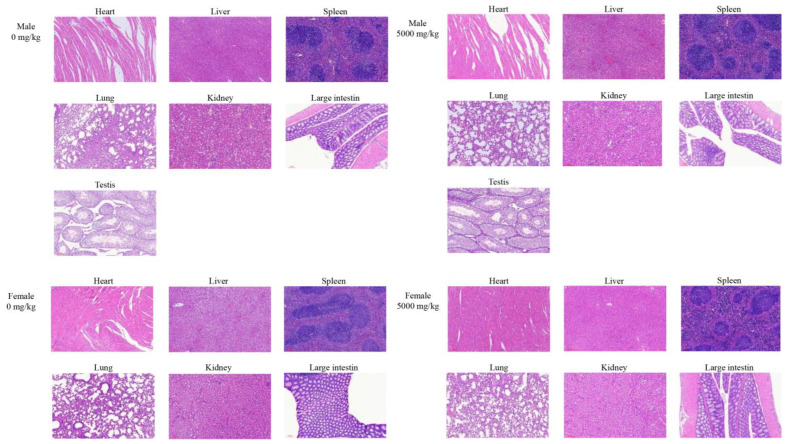
Histological examination. The male and female mice were fed with a fixed dose of 0 mg/kg BW and 5000 mg/kg BW of cMORP for the single-dose acute oral toxicity study (×100).

**Figure 8 foods-12-01590-f008:**
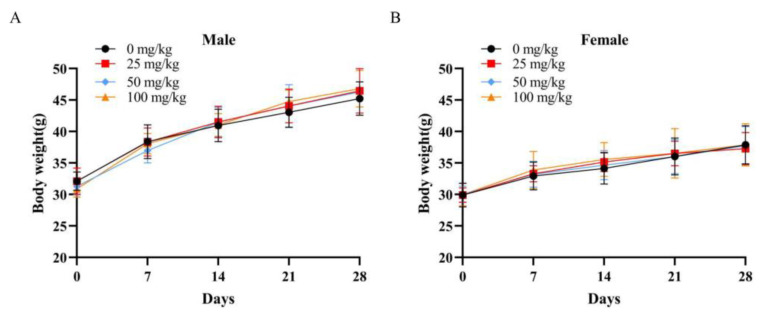
Body weight changes of mice in the repeated-dose 30-day oral toxicity study: (**A**) male; (**B**) female. Data are presented as mean ± SD (*n* = 10/sex/group).

**Figure 9 foods-12-01590-f009:**
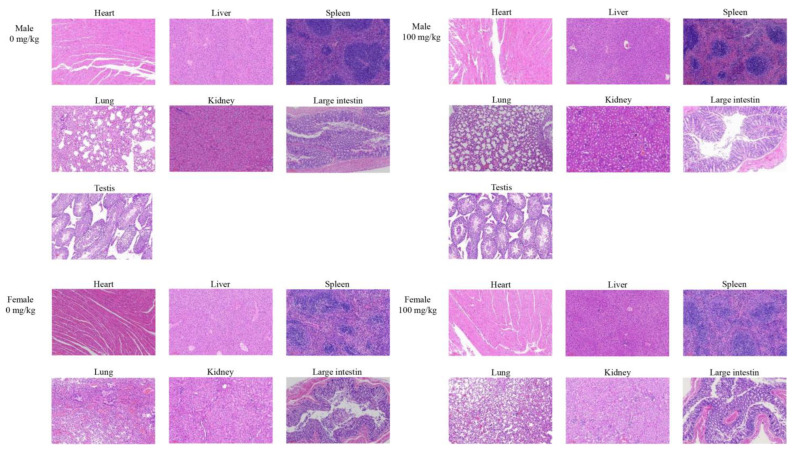
Histological examination. The male and female mice were fed with a fixed dose of 0 mg/kg BW and 100 mg/kg BW of cMORP for the repeated-dose 30-day oral toxicity study (×100).

**Table 1 foods-12-01590-t001:** Orthogonal design factor levels.

Level	Factors			
	A (Liquid/Solid Ratio)/mL·g^−1^	B (Temperature)/°C	C (Times)	D (Time)/h
1	15	70	1	0.5
2	20	80	2	1
3	25	90	3	2

**Table 2 foods-12-01590-t002:** The results of orthogonal experimental design.

Level			Factors		Yield (%)
	A	B	C	D	
1	1	1	1	1	7.31
2	1	2	2	2	8.06
3	1	3	3	3	4.60
4	2	1	2	3	7.48
5	2	2	3	1	7.75
6	2	3	1	2	4.30
7	3	1	3	2	7.01
8	3	2	1	3	8.52
9	3	3	2	1	4.20
K1	19.98	21.80	20.14	19.27	
K2	19.53	24.34	19.74	19.37	
K3	19.73	13.10	19.36	20.60	
k1	6.66	7.27	6.71	6.42	
k2	6.51	8.11	6.58	6.46	
k3	6.58	4.37	6.45	6.87	
R	0.15	3.75	0.26	0.44	

**Table 3 foods-12-01590-t003:** Process verification results.

Yield (%)	Average (%)	RSD (%)
8.59	8.37	2.44
8.32		
8.19		

**Table 4 foods-12-01590-t004:** Chemical composition of cMORP.

Samples	cMORP
Carbohydrate (%)	99.80 ± 1.52
Uronic acid (%)	<0.01
Protein (%)	<0.01

Data are represented as mean ± SD (*n* = 3).

**Table 5 foods-12-01590-t005:** Monosaccharide compositions of cMORP.

Samples	Percentage (%)
Ara	0.81
Gal	1.51
Glc	92.15
Rha	0.88
Gal-UA	1.98
Glc-UA	0.40
Gul-UA	2.28

**Table 6 foods-12-01590-t006:** Organ weights relative to body weight in the repeated-dose 30-day oral toxicity study.

Organs	0 mg/kg	25 mg/kg	50 mg/kg	100 mg/kg
Male				
Heart (%)	0.46 ± 0.05	0.49 ± 0.06	0.48 ± 0.05	0.48 ± 0.05
Liver (%)	4.37 ± 0.15	4.36 ± 0.26	4.34 ± 0.30	4.18 ± 0.32
Spleen (%)	0.30 ± 0.05	0.28 ± 0.06	0.32 ± 0.10	0.26 ± 0.07
Lung (%)	0.49 ± 0.07	0.51 ± 0.07	0.51 ± 0.03	0.49 ± 0.08
Kidneys (%)	1.33 ± 0.08	1.28 ± 0.06	1.27 ± 0.10	1.35 ± 0.12
Testes (%)	0.55 ± 0.07	0.54 ± 0.06	0.55 ± 0.08	0.56 ± 0.04
Female				
Heart (%)	0.42 ± 0.03	0.43 ± 0.05	0.43 ± 0.04	0.44 ± 0.05
Liver (%)	4.03 ± 0.64	4.33 ± 0.54	4.34 ± 0.52	4.10 ± 0.33
Spleen (%)	0.32 ± 0.09	0.34 ± 0.15	0.31 ± 0.07	0.31 ± 0.06
Lung (%)	0.62 ± 0.19	0.57 ± 0.14	0.62 ± 0.12	0.60 ± 0.09
Kidneys (%)	1.00 ± 0.11	1.03 ± 0.06	0.97 ± 0.08	0.99 ± 0.14

Units are given in brackets. Data are represented as mean ± SD (*n* = 10).

**Table 7 foods-12-01590-t007:** Hematological parameters in the repeated-dose 30-day oral toxicity study.

	0 mg/kg	25 mg/kg	50 mg/kg	100 mg/kg
Male				
WBC (10^9^/L)	4.94 ± 1.85	5.37 ± 1.48	6.06 ± 1.19	5.17 ± 2.30
RBC (10^12^/L)	10.49 ± 0.64	10.83 ± 0.46	10.53 ± 0.82	10.61 ± 0.91
HGB (g/L)	150.60 ± 6.00	151.30 ± 5.95	150.00 ± 10.98	151.70 ± 11.79
HCT (%)	44.21 ± 2.62	44.97 ± 1.75	44.92 ± 3.12	46.03 ± 3.94
MCV (fL)	42.21 ± 2.10	41.55 ± 1.53	42.72 ± 1.45	43.47 ± 2.61
MCH (pg)	14.39 ± 0.72	13.98 ± 0.33	14.25 ± 0.32	14.33 ± 0.63
MCHC (g/L)	341.20 ± 14.33	336.50 ± 6.67	333.80 ± 5.41	330.30 ± 19.33
PLT (10^9^/L)	1255.90 ± 539.08	1356.10 ± 300.27	1239.60 ± 292.11	1118.30 ± 298.78
PDW (fL)	8.40 ± 0.55	8.31 ± 0.53	8.04 ± 0.58	8.06 ± 0.64
MPV (fL)	7.24 ± 0.28	7.20 ± 0.34	7.11 ± 0.31	6.93 ± 0.52
P-LCR (%)	7.99 ± 1.69	7.75 ± 2.07	7.21 ± 1.53	6.88 ± 2.37
PCT	0.90 ± 0.37	0.97 ± 0.17	0.84 ± 0.30	0.74 ± 0.31
N (%)	17.15 ± 10.16	16.57 ± 3.80	20.91 ± 4.57	21.15 ± 4.21
L (%)	67.72 ± 13.98	69.47 ± 6.77	66.31 ± 8.14	66.58 ± 7.34
M (%)	13.64 ± 4.88	12.58 ± 5.44	10.95 ± 5.69	10.08 ± 6.57
E (%)	1.46 ± 0.55	1.37 ± 0.54	1.81 ± 0.42	2.16 ± 0.82
B (%)	ND	ND	ND	ND
Female				
WBC (10^9^/L)	4.63 ± 1.51	5.11 ± 1.65	4.96 ± 2.16	5.01 ± 2.38
RBC (10^12^/L)	10.93 ± 0.56	10.97 ± 0.48	10.82 ± 0.65	10.19 ± 0.80
HGB (g/L)	158.38 ± 5.88	155.70 ± 10.52	160.33 ± 7.42	150.00 ± 11.49
HCT (%)	46.54 ± 1.68	45.90 ± 2.15	47.14 ± 2.12	44.56 ± 2.25
MCV (fL)	42.66 ± 1.97	41.87 ± 0.92	43.66 ± 2.11	43.85 ± 2.64
MCH (pg)	14.51 ± 0.71	14.19 ± 0.69	14.83 ± 0.33	14.72 ± 0.52
MCHC (g/L)	340.25 ± 5.55	339.20 ± 13.93	340.11 ± 12.86	336.40 ± 16.49
PLT (10^9^/L)	1020.88 ± 171.38	1090.10 ± 254.49	927.00 ± 288.88	899.60 ± 323.82
PDW (fL)	8.49 ± 0.67	8.21 ± 0.61	7.91 ± 0.28	7.82 ± 0.59
MPV (fL)	7.18 ± 0.48	6.90 ± 0.37	6.93 ± 0.24	6.89 ± 0.45
P-LCR (%)	8.09 ± 2.61	6.29 ± 1.87	6.97 ± 0.79	6.47 ± 2.30
PCT	0.73 ± 0.13	0.75 ± 0.18	0.65 ± 0.22	0.62 ± 0.23
N (%)	17.04 ± 5.92	15.47 ± 6.38	16.10 ± 9.64	18.17 ± 5.33
L (%)	71.10 ± 9.22	72.50 ± 11.64	71.28 ± 17.80	70.12 ± 9.03
M (%)	10.33 ± 4.34	10.44 ± 6.76	10.64 ± 9.37	9.61 ± 6.88
E (%)	1.46 ± 0.61	1.57 ± 0.45	1.98 ± 1.67	2.09 ± 0.60
B (%)	ND	ND	ND	ND

Units are given in brackets. Data are represented as mean ± SD (*n* = 10). WBC, white blood cell count; RBC, red blood cell count; HGB, hemoglobin concentration; HCT, hematocrit; MCV, mean corpuscular volume; MCH, mean corpuscular hemoglobin; MCHC, mean corpuscular hemoglobin concentration; PLT, platelet count; PDW, platelet distribution width; MPV, mean platelet volume; P-LCR, platelet–larger cell ratio; PCT, procalcitonin; N, neutrophils percentage; M, monocytes percentage; L, lymphocytes percentage; E, eosinophils percentage; B, basophils percentage. ND means not detected.

**Table 8 foods-12-01590-t008:** Serum biochemical parameters in the repeated-dose 30-day oral toxicity study.

Parameters	0 mg/kg	25 mg/kg	50 mg/kg	100 mg/kg
Male				
ALT (IU/L)	46.70 ± 11.80	41.20 ± 7.97	43.50 ± 12.42	41.00 ± 11.76
AST (IU/L)	170.90 ± 36.21	158.80 ± 45.05	161.00 ± 39.39	161.60 ± 35.38
TP (g/L)	60.34 ± 3.50	59.89 ± 3.46	59.07 ± 4.14	60.42 ± 3.70
ALB (g/L)	35.93 ± 4.04	34.49 ± 2.23	34.79 ± 5.43	36.06 ± 3.10
GLU (mmol/L)	5.65 ± 0.94	5.77 ± 1.95	6.54 ± 2.14	6.61 ± 2.33
CR (mmol/L)	11.00 ± 5.33	11.70 ± 4.00	12.20 ± 4.52	11.90 ± 4.61
CHO (mmol/L)	3.39 ± 0.59	3.39 ± 0.37	3.38 ± 0.37	3.43 ± 0.35
TG (mmol/L)	2.00 ± 0.58	2.07 ± 0.91	2.04 ± 0.43	1.74 ± 0.37
Female				
ALT (IU/L)	42.10 ± 13.88	44.20 ± 15.53	41.70 ± 10.81	42.90 ± 12.30
AST (IU/L)	130.90 ± 18.31	127.70 ± 28.69	138.50 ± 14.01	124.60 ± 29.57
TP (g/L)	58.53 ± 2.48	58.85 ± 3.74	59.26 ± 3.82	58.58 ± 1.77
ALB (g/L)	35.57 ± 1.07	35.92 ± 1.73	36.56 ± 1.64	35.92 ± 1.08
GLU (mmol/L)	5.22 ± 0.95	5.45 ± 1.09	4.55 ± 1.13	5.26 ± 1.20
CR (mmol/L)	10.30 ± 4.03	12.80 ± 2.49	12.10 ± 4.25	11.50 ± 3.87
CHO (mmol/L)	2.70 ± 0.62	2.70 ± 0.33	2.79 ± 0.36	2.85 ± 0.60
TG (mmol/L)	1.91 ± 0.41	1.94 ± 0.43	1.65 ± 0.40	1.96 ± 0.55

Units are given in brackets. Data are represented as mean ± SD (*n* = 10). ALT, alanine aminotransferase; AST, aspartate aminotransferase; TP, total protein; ALB, albumin; GLU, glucose; CR, creatinine; CHO, cholesterol; TG, triglyceride.

## Data Availability

Data are contained within the article.

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
