# Peer review of "Optimization of Extraction Process, Preliminary Characterization and Safety Study of Crude Polysaccharides from Morindae Officinalis Radix"

_foods, 2023, doi:10.3390/foods12081590_

Round 1

Reviewer 1 Report

The manuscript "Optimization of Extraction Process, Preliminary Characterization and Safety Study of Crude Polysaccharides from Morindae Officinalis Radix" represent a very interesting and exhaustive research. In introduction part authors explained in detail the purpose and the aim of their research. Materials and methods part- research is well organized and designed, and materials and methods well described.

Results are correct presented,and discussion is supported with the obtained results.

Some minor changes are listed below: 

Line 13 - bracket "times of 1)" 

Line 105 - 107 - in explanation of calculation fomula is missing explanation for "Y". 

Line 348 - space after "liver, " 

Reviewer 2 Report

The manuscript works on the hot water extraction of crude polysaccharides from Morindae Officinalis Radix and performed some experiments on its toxicology. It was prepared scientifically , however, it requires major corrections.

Firstly, there is no optimization in the process of the extraction of polysaccharide, thereofore remove it or perform some experiments on the optimization.

secondly, the carbohydrates from the root is so high, more than 99%. Are you sure about it, without any purification, you obtained a pure carbohydrate. are you sure?

There are also some mistypos in the text which are required to correct and provided as follows:

Line 66: extraction times repeated twice.

Line 71: Scanning electron microscopy

Line 132: 1 ml polysaccharide solution or glucose solution at which concentration?

Line 148-150: at which resolution? Please indicate a reference for the FTIR. You can use this work “Effect of Thermal Treatment on Chemical Structure of Β-Lactoglobulin and Basil Seed Gum Mixture at Different States by ATR-FTIR Spectroscopy”

Reviewer 3 Report

the manuscript is quite interesting for the readers. several comments are addressed to the authors in order to enhance the quality of manuscript?

1. is it using subcritical water? 

2. add the parameters value in the abstract

3. add the a little explanation in the introduction related hot water extraction

4. add the objective of study in the last paragraph of introduction

5. where is explanation of  Y in the line 106

6.  extraction time is repetition? / reflux?

7.  why do you choose orthogonal design instead of other doe?

8. please add the reason why your doe is chosen in the results and discussion

9. add the responses in the table 1

10.  add the preliminary study related the chosen of parameter value

Round 2

Reviewer 2 Report

The authors replied to my comments and corrected the paper.